# Blood-Derived Exosomal hTERT mRNA in Patients with Lung Cancer: Characterization and Correlation with Response to Therapy

**DOI:** 10.3390/biomedicines11061730

**Published:** 2023-06-16

**Authors:** Ofer Rotem, Alona Zer, Lilach Yosef, Einat Beery, Hadar Goldvaser, Anna Gutkin, Ron Levin, Elizabeth Dudnik, Tamar Berger, Meora Feinmesser, Adva Levy-Barda, Meir Lahav, Pia Raanani, Orit Uziel

**Affiliations:** 1Davidoff Cancer Center, Rabin Medical Center, Petah Tikva 49100, Israel; oferro@clalit.org.il (O.R.);; 2Sackler Faculty of Medicine, Tel Aviv University, Tel Aviv 6997801, Israel; 3The Felsenstein Medical Research Center, Rabin Medical Center, Petah Tikva 49100, Israel; 4Shaare Zedek Medical Center, Faculty of Medicine, Hebrew University, Rehovot 7612001, Israel; 5Sheba Medical Center, Ramat Gan 5262000, Israel; 6Institute of Hematology, Rabin Medical Center, Petah Tikva 49100, Israel; 7Biobank, Department of Pathology, Rabin Medical Center, Petah Tikva 49100, Israel

**Keywords:** small-cell lung carcinoma (SCLC), non-small-cell lung carcinoma (NSCLC), hTERT, exosomes

## Abstract

**Background:** Telomerase (human telomerase reverse transcriptase (hTERT) is considered a hallmark of cancer, being active in cancer cells but repressed in human somatic cells. As such, it has the potential to serve as a valid cancer biomarker. Exosomal hTERT mRNA can be detected in the serum of patients with solid malignancies but not in healthy individuals. We sought to evaluate the feasibility of measuring serum exosomal hTERT transcripts levels in patients with lung cancer. **Methods:** A prospective analysis of exosomal hTERT mRNA levels was determined in serum-derived exosomes from 76 patients with stage III–IV lung cancer (11 SCLC and 65 NSCLC). An hTERT level above RQ = 1.2 was considered “detectable” according to a previous receiver operating characteristic curve (ROC) curve. Sequential measurements were obtained in 33 patients. Demographic and clinical data were collected retrospectively from patients’ charts. Data on response to systemic therapy (chemotherapy, immunotherapy, and tyrosine kinase inhibitors) were collected by the treating physicians. **Results:** hTERT was detected in 53% (40/76) of patients with lung cancer (89% of SCLC and 46% of NSLCC). The mean hTERT levels were 3.7 in all 76 patients, 5.87 in SCLC patients, and 3.62 in NSCLC patients. In total, 25 of 43 patients with sequential measurements had detectable levels of hTERT. The sequential exosomal hTERT mRNA levels reflected the clinical course in 23 of them. Decreases in hTERT levels were detected in 17 and 5 patients with partial and complete response, respectively. Eleven patients with a progressive disease had an increase in the level of exosomal hTERT, and seven with stable disease presented increases in its exosomal levels. Another patient who progressed on the first line of treatment and had a partial response to the second line of treatment exhibited an increase in exosomal hTERT mRNA levels during the progression and a decrease during the response. **Conclusions:** Exosomal hTERT mRNA levels are elevated in over half of patients with lung cancer. The potential association between hTERT levels and response to therapy suggests its utility as a promising cancer biomarker for response to therapy. This issue should be further explored in future studies.

## 1. Introduction

Lung cancer is the most often diagnosed malignancy in the world and the most frequent cause of cancer death. About 2.2 million new cases of lung cancer were diagnosed worldwide in 2021 [1]. It is estimated that 3 million patients with lung cancer will die by 2035 [2,3]. The prognosis for lung cancer is relatively poor, with a 5-year survival rate varying from 4% to 17%, depending on the stage of the disease at the time of diagnosis [3]. Unfortunately, 75% of patients are diagnosed at an advanced stage of the disease [4]. Although novel medications continue to improve outcome, prognosis remains poor, with a median overall survival (OS) of less than 3 years in most subpopulations of patients with advanced lung cancer [4]. The currently recommended screening for high-risk populations, a low-dose CT scan, improved early detection rates; however, the majority of the patients were still diagnosed with advanced disease [5]. Serum biomarkers have been investigated in lung cancer for two main purposes: early detection (in adjunct to CT screening) or as surrogate markers for response to therapy. The analysis of genomic epithelial serum tumor markers such as squamous cell carcinoma antigen (SCCA), carcinoembryonic antigen (CEA), or neuron-specific enolase (NSE) and cytokeratin 19 (CYFRA 21-1) [6], which have been reported to be elevated, may detect the disease only at advanced stages [7,8]. Other markers such as modulations of microsatellites, DNA hypermethylation in several genes (*BCAT1, CDO1, TRIM58, ZNF177*, and *CRYGD*), the mutational status of the *p53* and *KRAS* gene, and the expression of microRNA may improve early-stage diagnosis, but they are still under investigation [9]. Similarly, several other methods based on the analysis of volatile organic compounds and exhaled breath condensation analysis were also studied but are not yet implemented in clinical practice [10]. Due to these drawbacks, there is a need to search for other markers for early diagnosis for the effective treatment of the disease.

Telomeres, repetitive sequences located at the two ends of chromosomes, protect genome integrity by masking chromosomal ends from being recognized as double-strand breaks that need to be repaired. Telomeres gradually shorten with repeated cycles of cell division accompanied to DNA synthesis until they reach a threshold which signals the cells to stop dividing, undergo senescence, and die [11,12]. Telomerase (human Telomerase Reverse Transcriptase, hTERT) is a reverse transcriptase which maintains telomere lengths and thus prevents cellular senescence and cell death due to telomere shortening. In this way, telomerase provides dividing cells with a limitless lifespan. Because telomerase is absent in almost all human somatic cells but is ubiquitously expressed in more than 90% of cancer cells, it serves as an attractive biomarker that distinguishes normal and neoplastic cells [13]. Due to its specificity, mainly to cancer cells, and its essentiality for providing them with limitless replicative potential, telomerase was defined as a hallmark of cancer [14]. The role of telomerase activity and regulation was deeply studied in a plethora of cancers, and its activity was mostly correlated with the aggressiveness of the studied malignancies [15].

Lung cancer cells have been shown by many reports to be heavily dependent on telomerase activity both in vitro and ex vivo [16,17,18,19,20,21,22,23,24,25,26,27,28,29,30,31,32]. Moreover, similarly to other types of cancers, a positive correlation was described between the levels of telomerase activity or regulation and the aggressiveness of the disease [19,20,21]. Subsequently, studies were focused around using telomerase as a valid anti-cancer target in lung carcinoma. Importantly, several inhibitors of the enzyme were developed and tested, showing that the inhibition of the enzyme’s activity successfully eliminated lung cancer cells [32,33,34,35,36,37,38,39,40,41,42,43,44,45,46,47,48,49,50,51].

Telomerase activity is mainly regulated by the transcriptional level. In most studies that aimed at deciphering the regulatory mechanism of telomerase in various settings, the transcript of telomerase, hTERT, was found to be well correlated with the activity of the enzyme; therefore, in most cases, its levels are considered to reflect its activity [39,52].

Detecting the hTERT transcript in cells derived from the lung is not feasible and is only doable when biopsies are taken for clinical purposes. Therefore, non-invasive blood-sample-based liquid biopsies seem like the preferable solution for this problem.

One use for liquid biopsies is the analysis of the content of exosomes that originate from neoplastic cells and travel in the blood. There are three types of vesicles that are shredded from cells and circulate in the blood: exosomes, microparticles, and apoptotic bodies. In our study, we decided to use exosomes, the most investigated extracellular vesicles, as a liquid-biopsy-related approach to identify hTERT. Carrying a molecular cargo of various types of nucleic acids, proteins, and lipids, exosomes circulate in blood and other body fluids and may deliver their cargo to bystander cells [53]. Previously, we have shown that various types of cancer cells secrete exosomes that carry the hTERT mRNA transcript [54,55]. Whether the hTERT transcript is detected in exosomes from patients with lung carcinoma is currently unknown. Our study addressed the possible future use of exosomal hTERT as a diagnostic marker for lung carcinoma and for assessment response to systemic therapy.

## 2. Materials and Methods

### 2.1. Patients’ Characteristics

The clinical characteristics of the patients and the corresponding levels of exosomal hTERT mRNA are depicted in Table 1.

### 2.2. Study Population and Procedures

Patients with advanced lung cancer were enrolled prior to the initiation of anti-cancer therapy at the Davidoff Cancer Center at the Rabin Medical Center, Beilinson campus. After signing an informed consent form approved by the local IRB (Helsinki committee of Rabin Medical Center), 10 mL of blood in a serum separation tube was obtained from 81 healthy volunteers and 76 consecutive untreated cancer patients in the Davidoff Cancer Center. In total, 44 patients were followed during treatment to observe the kinetics of exosomally derived hTERT mRNA. The blood samples were centrifuged at 2500 R.P.M. for 10 min; the serum was collected, divided into 1 mL aliquots, and kept in −20 °C for exosomes and mRNA isolation. The hTERT levels obtained from 81 healthy volunteers served as controls.

Patients with more than one primary solid malignancy were excluded. Patients’ demographic and clinical data were extracted from the electronic medical records. The response to systemic anti-cancer therapy was assessed and categorized as complete response, partial response, stable disease, and progression of disease, as defined by the treating physicians.

### 2.3. Exosomes’ Purification

Exosomes were isolated from patients’ sera by using the Total Exosome Isolation Kit (Invitrogen, Carlsbad, CA, USA) according to the manufacturer’s instructions. The purity and concentration of exosomes were analyzed by using the NanoSight tracking device.

### 2.4. RNA Purification

RNA from exosomes was purified with a Total Exosome RNA and Protein Isolation Kit (Invitrogen, Waltham, MA, USA) according to the provided manual.

### 2.5. cDNA Formation

mRNA was reverse transcribed using the High-Capacity cDNA Reverse Transcription Kit (Applied Biosystems, Foster City, CA, USA) according to the manufacturer’s instructions.

### 2.6. hTERT Expression by Real-Time PCR

The expression of hTERT was measured relative to that of HPRT-1 as a reference gene. Gene amplification was executed using the following sets of primers (HyLabs, Rehovot, Israel).
hTERT: Forward, 5′-GTACTTTGTCAAGGTGGATGTGA-3′Reverse, 5′-GCTGGAGGTCTGTCAAGGTAGAG-3′.HPRT-1: Forward, 5′-TCAGGCAGTATAATCCAAAGATGGT-3′Reverse, 5′-CTTCGTGGGGTCCTTTTCAC-3′.

Polymerase chain reactions were prepared with the TaqMan fluorophore-labeled primers (Applied Biosystems, Waltham, MA, USA) and run and analyzed on the Step One Detection System (Applied Biosystems). Reactions were performed using 50 cycles; a normal value (no expression of hTERT) was arbitrarily defined as 1 for further calculation purposes.

### 2.7. Statistical Analysis

The statistical analysis was generated using the SAS Software, Version 9.4, 2002–2012 (SAS Institute Inc., Cary, NC, USA). Continuous variables were presented as mean ± std and median (minimum–maximum), and categorical variables were presented as (*n*, %). The normality of distribution for the continuous variables was assessed graphically and using a Kolmogorov–Smirnov test. If deemed normal, ANOVA (or *t*-test for two groups) was used to compare the value of continuous variables between study groups; if not deemed normal, Wilcoxon’s test was used. χ^2^ and/or Fisher’s exact test were used in the analysis of categorical variables between study groups. Pearson’s correlation was used to assess the association between continuous variables. Two-sided *p*-values <0.05 were considered statistically significant.

## 3. Results

### 3.1. Patients

In total, 76 patients were enrolled, 65 with NSCLC and 11 with SCLC. Out of those, 49 patients had adenocarcinoma, 17 had squamous cell carcinoma and 10 had large-cell carcinoma. Patients’ characteristics are presented in Table 1.

### 3.2. The Iolation of Exosomes

Basically, all living cells secrete vesicles, functioning as cell–cell communicators. There are several types of extra vesicles, differing in their cellular origin and biogenesis, size, release mechanism, molecular cargo, and function type. These include microvesicles (MVs), exosomes, and apoptotic bodies [53]. To show that our isolated vesicles are exosomes, we used NanoSight tracking analysis (NTA) and transmission electron microscopy (TEM). Based on Brownian motion, NTA identifies the size and concentration of vesicles of interest. The NTA results showed that a large concentration of the vesicles was at the expected size of exosomes (30–150 nm), and the TEM image indicated a similar size (Appendix A).

### 3.3. The Dynamics of hTERT at Time of Evaluation Regarding the Status of Metastasis

hTERT was hardly detected in the healthy control cohort (7.4%) but was detected in 53% (40/76) of patients with lung cancer (89% of SCLC and 46% of NSLCC). The difference between the average values of the control group and those of the cancer patient group was statistically significant (*p* = 2.2 × 10^−5^). The mean hTERT levels were 3.7 (±0.79) in all 76 patients, 5.87 (±1.6) in SCLC patients, and 3.62 (±0.29) in NSCLC with metastases (METs), large-cell neuroendocrine carcinomas (LCNECs), or sarcomatoid patients (Table 1, Figure 1). There were no differences between the NSCLC and the SCLC groups in this regard. The difference between the average values of the exosome-derived hTERT transcripts in patients with NSCLCs and those of patients with NSCLCs, LCNECs or sarcomatoids was statistically significant (*p* = 0.003).

No correlation was found between the extent of the disease and the levels of exosomal hTERT (*p* = 0.68). This lack of correlation between metastatic and non-metastatic disease suggests that the levels of exosomal hTERT cannot be differentiated between the two.

### 3.4. The Levels of Exosomal hTERT Transcript throughout Follow-Up of the Disease

We were able to follow 49 patients and measure the levels of exosomal hTERT transcripts in more than one sample throughout their disease. The results of this analysis are shown in Figure 2. As shown, the levels of the exosomal hTERT transcript decreased in most patients during the follow-up measurement, and the differences between the first measurement (at diagnosis) and the second one reached statistical significance (*p* = 0.047). Interestingly, out of 32 patients of which we had the clinical details regarding their response to treatment, in 23 of them (>72%), whether the patients were benefiting from therapy or not, the clinical outcome was in line with the levels of exosomal hTERT., e.g., when the response was defined as progressive disease, the levels of the exosomal hTERT transcript were increased. Two cases in which the dynamics of the levels of the exosomal hTERT transcript matched the clinical outcomes are presented in Figure 3.

The correlation between response to treatment and values of the hTERT transcripts in exosomes (Appendix A) approached clinical significance (*p* = 0.064).

## 4. Discussion

For the first time, our study shows the presence of the transcript of hTERT in exosomes derived from the sera of patients with lung cancer. The use of exosomes as diagnostic markers in early cancer detection, progression, response to treatment, and the identification of minimal residual disease (MDR) has been highly expanded in recent years [56]. As in several other types of malignancies, most patients diagnosed with primary lung carcinoma are already in the metastatic lethal stage of the disease upon the appearance of symptoms. Therefore, early diagnosis and treatment are critical to improve the survival of these patients.

As mentioned above, the current methodology of low-dose spiral CT is associated with a high rate of false positives and exposure to radiation and requires a solid biopsy of highly heterogenous tumors. To circumvent these associated drawbacks, liquid biopsies have been increasingly suggested as an alternative preferable diagnostic strategy, enabling the accurate diagnosis of lung cancer, the identification of the specific type of the malignancy, and the reveal of the mutational landscape of the tumor. Since multiple liquid biopsies can be taken throughout the course of the disease, the treatment of which is stage-dependent, patients’ stratification, the monitoring of responses to the disease treatments, and the detection of minimal residual disease post treatment are also possible, thus addressing precision oncology to improve clinical care [56].

The detection and analysis of circulating free DNA (cfDNA), circulating tumor cells (CTCs), and exosomes are the strategies used in most studies of liquid biopsies in cancer [57,58]. Traveling in the blood, cfDNA refers to DNA derived from all cell origins, whereas ctDNA only refers to DNA fragments originated from the tumor cells. As such, apart from DNA of genomic origin, cfDNA includes mitochondrial, fetal, extrachromosomal circular, and even microbial types of DNA ([56] and references therein). As the majority of cfDNA is of hematopoietic origin, some of which may contain benign clonal hematopoietic mutations which may be attributed to neoplastic cells, it is of high importance to differentiate between the two, based on their inherent differences. In the light of the sensitivity of the analysis of cfDNA, several studies reported impressive findings regarding the deciphering of the mutational makeup of lung cancer patients.

Interestingly, cfDNA levels in NSCLC patients are higher than those in patients with chronic respiratory inflammation or the healthy population, with very high specificity (90% and 80.5%, [59]), especially in those with stage II–IV NSCLC. Several studies were conducted comparing tissue to liquid-biopsy-derived analysis results. For example, a study of the ctDNA of cancer patients to identify mutations for relevant treatment (the TARGET study) demonstrated 78% similarity to mutations originated from the cognate tissues [60]. The detection of the EGFR mutation status, the most frequent driver mutation in NLSCS, using cfDNA analysis and the subsequent description of its positive association with the aggressiveness of the disease has been reported recently [61]. A similar study showed the relevance of this mutation upon the analysis of ctDNA [61,62]. In addition, numerous lung-cancer-related mutations (e.g., in *ALK, EGFR*, and *pmKRAS*) as well as the loss of heterozygosity, microsatellite instability, and gene methylation, are also readily detected in these cfDNA [62,63]. The recurrence and prognosis of lung carcinoma was also reported to be efficiently detected via cfDNA analysis, contributing to rationalized clinical decisions [64]. Along these lines, the identification of non-invasive early lung cancer in most pre-treated patients was achieved by using the deep sequencing of cfDNA, demonstrating its validity as a diagnostic lung cancer marker. Moreover, the somatic mutations found accurately reflected clonal hematopoiesis and were non-recurrent in these patients and in risk-matched controls [65]. The authors further developed a machine-learning-based platform for the accurate diagnosis of early-stage lung cancer using data obtained from cfDNA analysis [66].

The identification of alternative mutations in the *RGFR* gene in response to treatment was also reported (reviewed in ref. [56]). Other NSCLC-related mutations, including *MET, KRAS, RET, BRAF PI3K* assessed in response to treatments, were also detected successfully by analyzing cfDNA [67,68,69].

Another study successfully used a combination of both CTCs and cfDNA in the diagnosis of NSCLC [70].

Still, most studies point to the need for highly sensitive and standardized detection technologies for the valid widespread use of cfDNA analysis in clinical applications.

CTC counts correlated to tumor progression in patients with early lung adenocarcinoma and successfully detected typical mutant genes (*Notch1, IGF2, EGFR,* and *PTCH1*) [71,72] and genes in smokers as predictive biomarkers [73]. The precision treatment of patients with lung cancer was decided based on their tatus of the ALK gene mutation, detected via sequencing of their cognate CTC DNA content [74,75].

As for the cfDNA methodology, CTC detection still requires more effective enrichment and quantification methods.

As described in the introduction, exosome analysis has been reported as a diagnostic liquid-biopsy-based tool for lung carcinoma [53,54,55]. Most studies identifying exosomal cargo in plasma or sera derived from patients with lung carcinoma describe certain microRNA signatures as diagnostic markers [76,77,78,79,80,81,82]. All in all, the reported implication of exosomal cargo in the aggressiveness of lung cancer suggests that the inhibition of exosome formation and release may be a potential new strategy for the treatment of lung cancer [82]. Similarly to CTCs and cfDNA, the establishment and standardization of a reliable exosome extraction method is still needed for their use in clinical applications [83].

In our study, the levels of the hTERT transcript significantly differed between all lung cancer patients and the healthy control groups; however, a lack of correlation between these levels and disease staging was observed, suggesting that they do not correlate with the extent or burden.

These results are in contrast with other studies conducted in our laboratory showing that the exosomal hTERT transcript may be defined as a diagnostic marker. For example, in brain tumors, patients with glioblastoma multiforme (GBM) exhibited significantly higher levels of hTERT transcripts compared to patients with meningiomas, non-malignant brain tumors, or healthy controls (submitted). Likewise, the levels of the hTERT transcript were correlated with levels of carcinoembryonic antigen (PSA) in patients with colorectal cancer and significantly differed from those of patients with polyps that served as a control group. In this study, we also measured the levels of the hTERT transcript in patients with Lynch syndrome, which is associated with a predisposition to colorectal cancer, and found positive levels of hTERT transcripts in several carriers [84].

Although they did not solely address the levels of hTERT exosomal transcripts in lung cancer, a few recent studies demonstrated the use of lung-derived exosomal content as diagnostic, prognostic, and potentially therapeutic agents [85,86]. A relevant study demonstrated that tumor-associated exosomes promoted the aggressiveness of lung cancer by increasing metastasis formation [87].

Of note, the engulfment of nucleic acids and other molecules in exosomes is actively mediated, mainly by the endosomal sorting complex required for transport (ESCRT) protein complex. This engulfment of certain molecules may affect their concentration in exosomes, which may differ from their original expression in the mother cells [88]. This ESCRT-mediated process is only partially elucidated [89] and may differ among individual patients, affecting the expression of certain mRNAs such as the hTERT in exosomes. In addition, the correlation between the activity of telomerase and the burden of disease in lung cancer is controversial. Whereas many studies reported a positive correlation between the activity of the enzyme and disease progression [32,33,34,35,36,37,38,39,40,41,42,43,44,45,46,47,48,49,50,51], one study showed the opposite: patients with low telomerase activity in their mononuclear cells were at a significantly increased risk of lung cancer compared to patients with high telomerase activity [90]. Another recent study showed that liquid biopsies based on the detection of circulating tumor cells’ hTERT can effectively diagnose pulmonary nodules to improve efficient CT diagnosis in patients with lung cancer [91]. Telomerase-positive circulating tumor cells were shown to be associated with poor prognosis in glioma [92].

We have shown that CTC capturing based on membranal telomerase-derived peptides both in solid and hematological malignancies is feasible [93]. In another study, the inhibition of telomerase in lung cancer cell lines suppressed their growth by negatively regulating telomerase expression [94]. Along these lines, the ectopic expression of hTERT promoted epithelial–mesenchymal transition in lung cancer cells [95].

This study bears a major limitation, which is related to the number of analyzed patients. Since this was a pilot study, the number of patients was not large enough to draw more solid conclusions.

All in all, it seems that a differential expression of the transcript exists in different pathological cases and biological setting. The combination assessments of other markers, preferentially in exosomes representing efficient liquid biopsies, will probably shed more light on defining valid diagnostic markers for upregulating the treatment of patients with lung cancer.

## Figures and Tables

**Figure 1 biomedicines-11-01730-f001:**
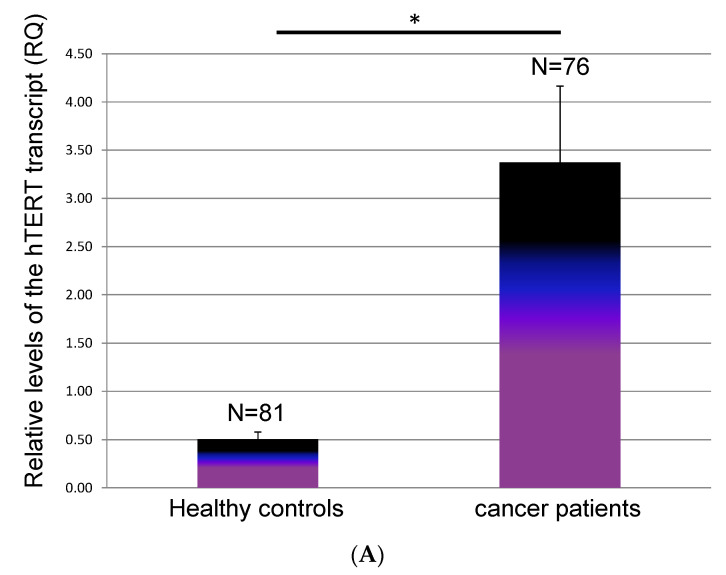
Levels of the exosomal hTERT transcript in patients with lung cancer. (**A**) Average values. (**B**) % of positive hTERT transcripts in patients with lung cancer. (**C**) Average values of exosomal hTERT in different lung malignancies; (**D**) % of positive hTERT transcripts in patients with different types of lung cancer. * denotes *p* < 0.05.

**Figure 2 biomedicines-11-01730-f002:**
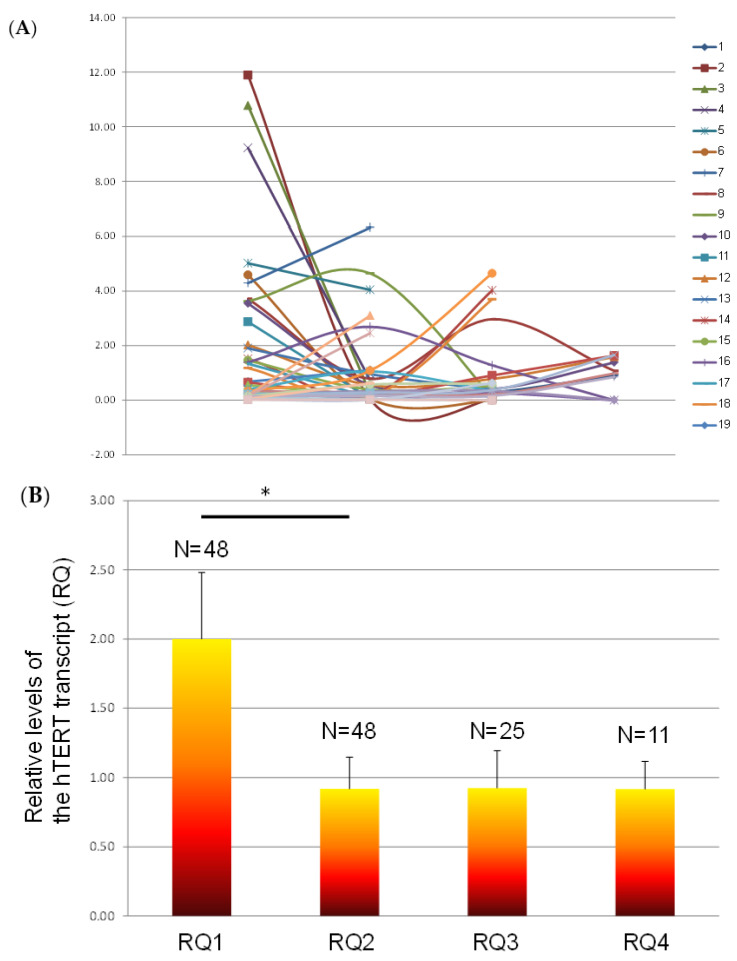
Levels of the exosomal hTERT transcript in longitudinal follow-up during 2–4 measurements. (**A**) Values of individual patients. (**B**) Average values. * Denotes *p* < 0.05.

**Figure 3 biomedicines-11-01730-f003:**
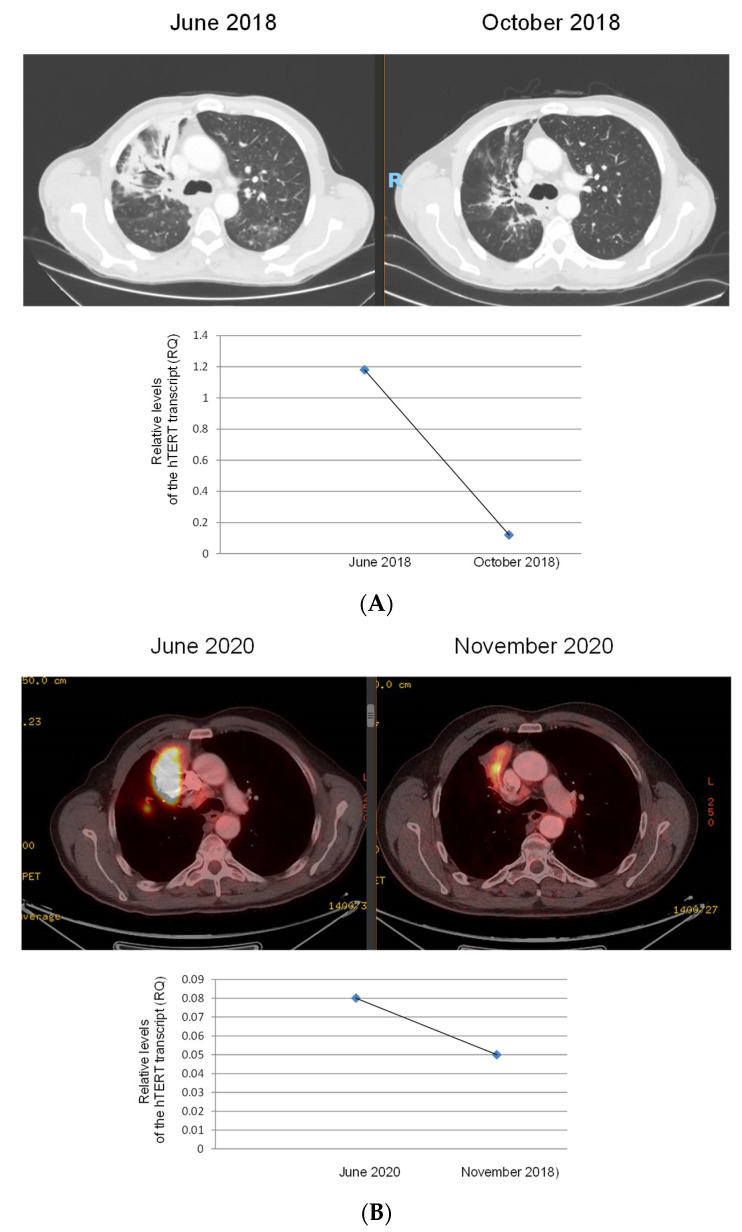
Imaging analysis of two patients during follow-up and the corresponding values of exosomal hTERT transcript. (**A**) CT imaging over 4 months (patient #1); (**B**) PET CT imaging over 5 months (patient #2).

**Table 1 biomedicines-11-01730-t001:** Patients’ characteristics and the levels of exosomal hTERT mRNA.

		Age		Gender		RQ					
	*n*	Median	Range	F	M	Mean RQ	SEM	Median	Range	Type	Stage
positives	40	66.5	46–85	28	12	6.45	1.44	3.67	1.23–44.61	NSCLC-30, SCLC-10	IIIa-2, IIIb-3, IV-35
negatives	36	67	38–88	22	14	0.36	0.05	0.33	0–1.18	NSCLC-30, SCLC-3, others-3	IIIb-10, IV-26

Positives: exosomal hTERT levels ≥ 1.2; negatives: exosomal hTERT < 1.2, SEM—standard error mean, RQ—relative quantitation of hTERT levels.

## Data Availability

Not applicable.

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
