# Peer review of "Blood-Derived Exosomal hTERT mRNA in Patients with Lung Cancer: Characterization and Correlation with Response to Therapy"

_biomedicines, 2023, doi:10.3390/biomedicines11061730_

Round 1

Reviewer 1 Report

1.This research focused on Blood-derived exosomal hTERT mRNA in patients with lung cancer: characterization and correlation with response to therapy, after check the pubmed,no related references, this manuscript was very prospective and significant.

2.Not all lung cancer patients revealed hTERT mRNA positive, Indicating poor specificity, how can we use this for Clinical medicine?

3. Can hTERT mRNA  and other gene combined for the diagnosis of late stage lung cancer?

4. Figures can be much more pefect, suh as I don't think it's necessary to use color for the bar chart .

Author Response

1.This research focused on Blood-derived exosomal hTERT mRNA in patients with lung cancer: characterization and correlation with response to therapy, after checking the PubMed, no related references, this manuscript was very prospective and significant.

We thank the reviewer for this worm response.

2.Not all lung cancer patients revealed hTERT mRNA positive, indicating poor specificity, how can we use this for Clinical medicine?

Absolutely correct! Therefore, we suggest using it in combination with other markers as said at the end of the discussion, line 303.

  1. Can hTERT mRNA and other gene combined for the diagnosis of late-stage lung cancer?

I do hope so....other studies are needed to reach a stable conclusion regarding this issue...

  1. Figures can be much more perfect, such as I don't think it's necessary to use color for the bar chart.

This may be right but since its cost is the same, we do prefer colourful bars... hope the reviewer understands this point.

All and all we highly appreciate the reviewer remarks for our manuscript!

Reviewer 2 Report

This paper suggests that exosomal hTERT RNA is promising marker for lung cancer.

This is a valuable report, however it needs to be double-checked to reduce the number of errors and inconsistencies.

Abstract

line 28: what it is the “level above 1.2”?

line 41: it is not appropriate to indicate % in the Conclusions section, because this study did not aim at precise estimation of the proportion of the marker-positive patients.

Results

There are too many sections, please merge some of them. I also suggest to move some Tables and Figures to the Supplement.

Section 3.2 needs more explanation.

Discussion

Please compare exosomal hTERT RNA against other techniques of the liquid biopsy in lung carcinomas.

Abstract/Introduction/Results/Discussion

There are multiple inaccuracies in the text (erroneous use of capital letters; missing spaces between words and symbols; wrong formatting; avoidable mistakes in English; lack of consistency while describing numerical results of the study, etc.). Please improve significantly the quality of the manuscript presentation if the Editors will offer a chance to revise the paper.  

The authors look proficient in English, but the text has too many avoidable mistakes.  

Author Response

This paper suggests that exosomal hTERT RNA is promising marker for lung cancer.

This is a valuable report, however it needs to be double-checked to reduce the number of errors and inconsistencies.

We thank the kind reviewer for its encouraging response!

Abstract

line 28: what it is the “level above 1.2”?

Correct, RQ=1.2 was added.

line 41: it is not appropriate to indicate % in the Conclusions section, because this study did not aim at precise estimation of the proportion of the marker-positive patients.

OK, this has now changed into: in over half of patients…

Results

There are too many sections, please merge some of them. I also suggest to move some

Absolutely, three sub-sections are now merged and called: “The dynamics of hTERT at time of evaluation, regarding the status of metastasis and in response to treatment”.

Tables and Figures to the Supplement.

Section 3.2 needs more explanation.

Correct. This section was expanded and re-written as follows:Basically, all living cells secrete vesicles, functioning as cell- cell communicators. There are several types of extra vesicles, differing in their cellular origin and biogenesis, size, release mechanism, molecular cargo and function type. These include microvesicles (MVs), exosomes, and apoptotic bodies [53]. To show that our isolated vesicles are exosomes, we have used the NanoSight tracking analysis (NTA) and transmission electron microscopy (TEM). Based on Brownian motion, NTA identifies the size and concentration of vesicles of interest. The NTA results showed that a large concentration of the vesicles was at the expected size of exosomes (30-150 nm) and the TEM image indicated a similar size (Fig. 1).

 Discussion

Please compare exosomal hTERT RNA against other techniques of the liquid biopsy in lung carcinomas.

Again- a very good remark. We have added the following to the beginning of the discussion: “Our study shows for the first time the presence of the transcript of hTERT in exosomes de-rived from the sera of patients with lung cancer. The use of exosomes as diagnostic marker in early cancer detection, progression, response to treatment and identification of minimal residual disease (MDR) has been highly expanded in recent years [56]. As in several other types of malignancies, most patients diagnosed with primary lung carcinoma are already in the metastatic lethal stage of the disease upon the appearance of symptoms. Therefore, early diagnosis and treatment are critical to improve the survival of these patients. As mentioned above, the current methodology of low dose spiral CT is associated with a high rate of false positive and exposure to radiation and requires solid biopsy of highly heterogenous tumors. To circumvent tumor heterogeneity and the above- mention drawbacks of solid biopsies, liquid biopsies has been increasingly suggested as an alternative preferable diagnostic strategy, enabling not only the diagnosis of lung cancer but also the specific type of the malignancy and the mutational landscape of the tumor, thus improving clinical care. The most studied liquid biopsy types detect and analyse circulating tumor cells (CTCs), circulating tumor DNA (cfDNA) and exosomes. [56, 57]. Based on immune enrichment, numerous studies have shown promising results in using CTCs as diagnostic markers. CTC counts correlated to tumor progression in patients with early lung adenocarcinoma and successfully detected typical mutant genes (Notch1, IGF2, EGFR, and PTCH1) [58, 59] and in smokers as predictive biomarkers [60]. Precision treatment of patients with lung cancer was decided based on their tatus of the ALK gene mutation, detected by sequencing of their cognate CTCs [61- 63]. Although promising, CTC detection still requires a more effective enrichment and quantification methods.

   Because the level of Cell-free DNA (cfDNA) increases upon carcinogenesis [64, 65], their isolation is relatively easy and efficient and due to their correlation with tumor load (at least in single patient), they are considered good diagnostic markers [66]. Interestingly, their level in NSCLC patients is higher than that of patients with chronic respiratory inflammation or the healthy population, with a very high specificity (90% and 80.5%, [67]), especially in with stage II–IV NSCLC. In addition, numerous lung cancer related mutations (e.g. in ALK, EGFR, pmKRAS) as well as loss of heterozygosity, microsatellite instability, and gene methylation are also readily detected in these cfDNA [68, 69]. Recurrence and prognosis of lung carcinoma was also reported to be efficiently detected by cfDNA analysis [70-74], contributing to rationalized clinical decisions [75]. Still, highly sensitive and standardised detection technologies are needed for a valid widespread clinical ap-plication [76]. 

   As described in the introduction, exosome analysis has been reported as a diagnostic liquid biopsy- based tool for lung carcinoma [53-55]. Most studies identifying exosomal cargo in plasma or sera derived from patients with lung carcinoma describe certain microRNA signature as diagnostic markers [77- 83]. All in all, the reported implication of exosomal cargo in the aggressiveness of lung cancer suggests that the inhibition of exosome formation and release may be a potential new strategy for the treatment of lung cancer [83]. Similarly, to CTCs and cfDNA, the establishment and standardisation of a reliable exosome extraction method is still needed for using them in clinical applications [84]”.

Abstract/Introduction/Results/Discussion

   There are multiple inaccuracies in the text (erroneous use of capital letters; missing spaces between words and symbols; wrong formatting; avoidable mistakes in English; lack of consistency while describing numerical results of the study, etc.). Please improve significantly the quality of the manuscript presentation if the Editors will offer a chance to revise the paper.

We do apologize for the multiple inaccuracies. The text has been revised by a scientific editor and to the best of our knowledge all these inaccuracies were corrected.

Comments on the Quality of English Language

The authors look proficient in English, but the text has too many avoidable mistakes. 

We deeply acknowledge the second reviewer for his careful reading of our manuscript and for raising smart points, which upon taking care of, highly improved our manuscript.  

Round 2

Reviewer 1 Report

As now we checked the revised manusrcipt,the authors have answered all my concerns,so I hope can be accepted by this famous jounal,the final decison only can be made by the editor in chief.

Author Response

We thank the reviewer for his coments!

Reviewer 2 Report

The authors did not manage to avoid inaccuracies. For example, line 274 cites refs 61-63 in the context of ALK translocations, however these papers are devoted to another topic.

The authors did not consider the suggestion to move some Figures and Tables into the Supplement.

The paragraph devoted to the cfDNA (lines 276-287) does not describe this field properly. I suggest the authors read the extensive literature on the detection of mutated cfDNA in lung cancer patients and address this topic in a better way.

Author Response

Answers for the 2nd reviewer

  1. The authors did not manage to avoid inaccuracies. For example, line 274 cites refs 61-63 in the context of ALK translocations, however these papers are devoted to another topic.

Thank you, we apologize for these leftover mistakes, which have been corrected now.

  1. The authors did not consider the suggestion to move some Figures and Tables into the Supplement.

We have moved Fig. 1 to the supplemented materials (as Suppl. Fig. 1) and Table 2 (as Suppl. Table 1) from the main text of the manuscript and changed all fig’s and tables references in the text as well.

  1. The paragraph devoted to the cfDNA (lines 276-287) does not describe this field properly. I suggest the authors read the extensive literature on the detection of mutated cfDNA in lung cancer patients and address this topic in a better way.

OK, has been done by re writing the following paragraph:

     As mentioned above, the current methodology of low dose spiral CT is associated with a high rate of false positive and exposure to radiation and requires solid biopsy of highly heterogenous tumors. To circumvent these associated drawbacks, liquid biopsies has been increasingly suggested as an alternative preferable diagnostic strategy, enabling the accurate diagnosis of lung cancer, the identification of the specific type of the malignancy and revealing the mutational landscape of the tumor. Since multiple liquid biopsies can be taken throughout the course of the disease of which treatment is stage dependent, patients’ stratification, monitoring response to the disease treatments and the detection of minimal residual disease post treatment are also possible, thus addressing precision oncology to improve clinical care [56].

   The detection and analysis of circulating free DNA (cfDNA), circulating tumor cells (CTCs), and exosomes are the most studies strategies of liquid biopsies in cancer [57, 58]. Travelling in the blood, cfDNA refers to DNA derived from all cell origins whereas ctDNA refers only to DNA fragments originated from the tumor cells. As such, apart of DNA of genomic origin, cfDNA includes mitochondrial, fetal, extrachromosomal circular and even microbial types of DNA [56 and references therein]. As the majority of cfDNA is of hematopoietic origin, some of which may contain benign clonal hematopoietic mutations which may be attributed to the neoplastic cells, it is of highly importance to differentiate between the two, based on their inherent differences. In the light of the sensitivity of analysis of cfDNA, several studies reported impressive findings regarding the deciphering of the mutational makeup of lung cancer patients.

Interestingly, cfDNA levels in NSCLC patients is higher than that of patients with chronic respiratory inflammation or the healthy population, with a very high specificity (90% and 80.5%, [59]), especially in with stage II–IV NSCLC. Several studies were conducted comparing tissue to liquid biopsies derived analysis results. For example, ctDNA of cancer patients to identify mutations for relevant treatment (the TARGET study) demonstrated 78% similarity to mutations originated from the cognate tissues [60]. The detection of the EGFR mutation status, the most frequent driver mutation in NLSCS, using cfDNA analysis and the subsequent description of its positive association with the aggressiveness of the disease has been reported recently [61]. A similar study showed the relevant of this mutation upon analysis of ctDNA [61, 62]. In addition, numerous lung cancer related mutations (e.g. in ALK, EGFR, pmKRAS) as well as loss of heterozygosity, microsatellite instability, and gene methylation are also readily detected in these cfDNA [62, 63]. Recurrence and prognosis of lung carcinoma was also reported to be efficiently detected by cfDNA analysis, contributing to rationalized clinical decisions [64]. Along these lines, the identification of non-invasive early lung cancer in most pre- treated patients was achieved by using deep sequencing of cfDNA, demonstrating its validity as a diagnostic lung cancer marker. Moreover, the somatic mutations found accurately reflected clonal haematopoiesis and were non-recurrent in these patients and in risk- matched controls [65]. The authors further developed a machine learning based platform for the accurate diagnosis of early- stage lung cancer using data obtained from cfDNA analysis [66].

The identification of alternative mutations in the RGFR gene in response to treatment was also reported [reviewed in ref. 56]. Other NSCLC related mutations including in MET, KRAS, RET, BRAF PI3K assessed in response to treatments were also detected successfully by analyzing cfDNA [67-69]. 

Another study successfully used a combination of both CTCs and cfDNA in the diagnosis of NSCLC [70].

Still, most studies point to the need for highly sensitive and standardized detection technologies for a valid widespread of cfDNA analysis for clinical application.

   CTC counts correlated to tumor progression in patients with early lung adenocarcinoma and successfully detected typical mutant genes (Notch1, IGF2, EGFR, and PTCH1) [71,72] and in smokers as predictive biomarkers [73]. Precision treatment of patients with lung cancer was decided based on their tatus of the ALK gene mutation, detected by sequencing of their cognate CTCs DNA content [74,75].

   As for the cfDNA methodology, CTC detection still requires a more effective enrichment and quantification methods.

References have been corrected accordingly.

We highly appreciate the reviewer thorough reading of our manuscript and his valuable suggestions!

Round 3

Reviewer 2 Report

The content of the revised version of the paper is good. However, the numbering of newly inserted references appears to be confused and their formatting is not uniformly adjusted to the MDPI journal style.